# Risk assessment of workers' exposure to BTEX and hazardous area classification at gasoline stations

**Sunisa Chaiklieng** *

Department of Environmental Health, Occupational Health and Safety, Faculty of Public Health, Khon Kaen University, Khon Kaen, Thailand

* csunis@kku.ac.th

**Data Availability Statement:** All relevant data are within the manuscript and its Supporting Information files.

**Funding:** This study was partially supported by research funding from The National Research

## Abstract

Vaporization of benzene, toluene, ethylbenzene, and xylene (BTEX) compounds pollutes the air and causes health hazards at gasoline stations. This study revealed the risk of BTEX exposure according to the hazardous area classification at gasoline stations. The risk assessment of gasoline workers from a representative group of 47 stations, which followed the United States Environmental Protection Agency-IRIS method of assessing BTEX exposure, was expressed as the hazard index (HI). A result of matrix multipliers of the hazardous exposure index and fire possibility from flammable gas classified hazardous area-I and area-II at the fuel dispensers. BTEX concentrations were actively sampled in ambient air and a flammable gas detector was used to measure the flammability level. Results showed that the BTEX concentrations from ambient air monitoring were in the range of 0.1–136.9, 8.1–406.0, 0.8–24.1 and 0.4–105.5 ppb for benzene, toluene, ethylbenzene, and xylene, respectively, which exceeded the NIOSH exposure limit of 100 ppb of benzene concentration. The risk assessment indicated that five stations reached an unacceptable risk of worker exposure to BTEX (HI>1), which correlated with the numbers of gasoline dispensers and daily gasoline sold. The risk matrix classified hazardous area-I at 4 meters and hazardous area-II at 4–8 meters in radius around the fuel dispensers. This study revealed the hazardous areas at gasoline stations and suggests that entrepreneurs must strictly control the safety operation practice of workers, install vapor recovery systems on dispenser nozzles to control BTEX vaporization and keep the hazardous areas clear of fire ignition sources within an eight-meter radius of the dispensers.

## Introduction

Exhaust fumes from refueling vehicles at gasoline stations lead to a poor quality of ambient air due to volatile organic solvent evaporation of BTEX compounds (benzene, toluene, ethylbenzene, and xylene). A previous report confirmed that the annual average concentration of volatile organic compounds (VOCs) from fuel evaporation of oil and gas stations in Northwest China was 21.5±1.0% [1]. That is one reason for the induction of hazardous conditions with

Council of Thailand (NRCT610007). No additional external funding received for this study.

**Competing interests:** The author have declared that no competing interests exist.

contributions from fuel vapor pollution [2]. Gasoline workers and clients exposed to BTEX while refueling are affected by adverse health effects. Benzene is a human carcinogen which also affects the human hematopoietic and reproductive systems [3]. Ethylbenzene and xylene have caused abnormal respiratory and neurological effects [4]. Gasoline workers had health risks from BTEX chronic exposure and some workers had a higher than acceptable risk of BTEX exposure [5, 6]. It was also shown that the lifetime cancer risk of exposure to benzene in Thai gasoline workers exceeded the acceptable limit [5, 7, 8] and this risk was similar to that found in the studies in Malaysia [9, 10]. The health risks of working in hazardous zones at gasoline stations related to BTEX exposure are still not clear with regard to the contribution from sources of vaporization.

In Thailand, maximum concentrations of 222, 527, 26 and 114 ppb were found for benzene, toluene, ethylbenzene, and xylene, respectively, at the gasoline stations of Bangkok [11]. The mean concentration of benzene was found to be 45.36 ppb at the gasoline stations in the northeast of Thailand [12]. With regard to other countries [13, 14], the concentrations found there were lower compared to Thai gasoline stations, such as in the gasoline stations of Brazil, where BTEX concentrations were 9.30, 12.7, 5.4 and 11.42 ppb for benzene, toluene, ethylbenzene and xylene, respectively [14]. Gasoline station workers in Kuwait were exposed to concentrations in excess of the NIOSH exposure limits [15]. The number of gasoline stations has risen by an average of 4–5 percent a year to 28,753 stations in Thailand [16] and they have been controlled by the Ministry of Energy. The level of benzene in fuel was reduced from 3.5% by volume to 1% by volume in 2012 [17]. Vapor recovery systems (VRS) have been installed at gasoline stations in Bangkok and its surrounding areas since 2007, according to a policy that did not cover other regions of Thailand.

The hazardous area classification related to human health risk and fire risk at gasoline stations was not adequately studied in the previous research. This study aimed to classify complex hazardous areas according to the health risk indications of gasoline workers and flammable gas evaporation during the refueling of vehicles.

## Materials and methods

### Sample size

This study was conducted at 47 gasoline stations. The representative sample size of the gasoline stations was calculated using Cochran [18] under the known number of a total of 64 stations located on both sides of Mittraphap Road, the main highway in Khon Kaen province, Thailand [16]. There were three inclusion criteria for the gasoline stations: 1) they had to be located in an area along either side of or within 5 km of Mittraphap Road, the main highway connecting Khon Kaen province to Bangkok, the capital city of Thailand, 2) they had to have more than eight dispenser nozzles and gasoline dispenser nozzles; and 3) they had to have more than 8 hours of daily operation time. This study was approved by Khon Kaen University's Ethics Committee for Human Research (No. HE612102). All participants from the gasoline stations gave written informed consent prior to taking part in the study and workers aged under 18 years were not included.

### Data collection

A gasoline station survey form was used to collect the characteristics of 47 gasoline stations and answers to questions regarding workers' characteristics, i.e. working hours per day and number of work days per week, were collected from the fueling workers. The BTEX concentrations were measured by active sampling with a charcoal tube at a flow rate of 0.2 L/min, as described in NIOSH method 1501 [19]. The exposure to BTEX was monitored during each

working shift; this monitoring covered the peak service hours for 4 hours of sampling and all charcoal tubes were kept at 4˚C while being transported to the laboratory. Each tube was extracted with carbon disulfide ($CS_2$), and analyzed by gas chromatography (GC) with flame ionization detector (FID) (Varian, CP3800), CP 52 wax column (30 m x 0.25 μm x 0.25 mm) of Hewlett Packard 1996, Germany, and a limit of detection (LOD) which was < 0.001 ppm and < 0.003 ppb for toluene and benzene and < 0.05 ppb for ethyl- benzene and xylene. The results of the BTEX concentrations were used to estimate the human health risk of exposure to BTEX.

A flammable gas detector was used to measure the total emission of VOCs with a photo ionization detector (PID) sensor (detection range of 0 to 1000 ppm) and a lower explosive limit—upper explosive limit (LEL-UEL) with Non-Dispersive Infrared (NDIR) combustible sensors (range of 0–100%LEL) at a distance of 0.15 meters from the fuel emission point during the refueling of vehicles. The wind velocity average was recorded on the day of BTEX collection and VOCs and flammable gas measurements.

The samples were collected between June and July 2018, at times when the temperature was 28.08 ±1.58˚C, humidity was 78.51±8.89%, and wind velocity was 10.0 ±2.82 m/min.

## Human health risk assessment

The health risk assessment for non-cancer effects was calculated by using the hazard quotient (HQ), which followed the United States Environmental Protection Agency-Integrated Risk Information System (EPA-IRIS) [20] according to $HQ_{BTEX}$ = EC/RfC, where EC represents exposure concentration via inhalation intake in μg/m³, and RfC is the reference concentration of daily exposure to BTEX in μg/m³ (30 μg/m³ of benzene, 5,000 μg/m³ of toluene, 1,000 μg/m³ of ethylbenzene, and 100 μg/m³ of xylene) [21, 22]. A resulting HQ of ≥1 was considered as "adverse non-cancer health effects of concern," and a HQ of <1 was considered an "acceptable level". The health risk was expressed as the hazard index (HI), which was determined by the summation of all HQ values of BTEX in the same area (HI = $\Sigma HQ_{BTEX}$). The exposure level (EC) was calculated by following USEPA [22] as in the formula of EC = (CA x ET x EF x ED) / AT, where CA represents contaminant concentration in μg/m³, EF represents exposure frequency in days/year (324 days/year; 6 days/week x 54 weeks/year from on-site interviews), ET represents exposure time in hours/day (9–11 hours/day from on-site interviews), ED represents exposure duration (25 years; 324 days/year) and AT represents average time (219,000 hours; 25 year x 365 days/year x 24 hour/day).

## Hazardous area classification at gasoline stations

Hazardous area (HZ) classification was determined according to emission of VOCs from the result of matrix multipliers of the hazardous exposure index (HzI) and fire possibility (flammability limit; %LEL-UEL) based on exposure which simulated distance from a source along the horizontal plane in the wind direction in meters (m) (x; distance 0.15–10 m). OSHA [23] and the AS/NZS 4360:2004 risk management standard [24] were applied as per the following equation and matrix table (Table 1);

$$HZ = HzI \text{ x } P$$

where HZ represents hazardous area, HzI represents the subsequent hazardous exposure index (4 levels, 4 scores) and P represents possibility (flammability limit level; 2 levels; 2 scores). The hazardous area (HZ) was divided into two areas, i.e. hazardous area-I and hazardous area-II. The scores of hazardous $BTEX_{emission}$ exposure risk were calculated according to

**Table 1. Risk matrix multipliers for hazardous area classification (area I, II).**

| Hazardous exposure index (HzI) | Flammability limit (%LEL-UEL) | |
| --- | --- | --- |
| | **Out-range of 1.3–7.4% (1)** | **In-range of 1.3–7.4% (2)** |
| HzI >2 (4) | hazardous area-I (score of 4) | hazardous area-I (score of 8) |
| HzI = >1–2 (3) | hazardous area-I (score of 3) | hazardous area-I (score of 6) |
| HzI = 0.5–1.0 (2) | hazardous area-II (score of 2) | hazardous area-I (score of 4) |
| HzI <0.5 (1) | hazardous area–II (score of 1) | hazardous area-II (score of 2) |

distance from the point sources and BTEX$_{emission}$ (BTEX$_{emiss}$) concentration was converted from the total concentration of VOCs.

The criteria of the hazardous areas were estimated from the results of the risk assessment scores (out of 8 points), which were divided into two groups: a score of 3–8, which was classified as "hazardous area-I" and a score of 1–2, which was classified as "hazardous area-II", as shown in Table 1.

The hazardous exposure index (HzI) was calculated as the summary hazard quotient (HQ) of BTEX$_{emiss}$ exposure according to the equation of HzI = ΣHQ$_{BTEXemiss}$. The hazardous exposure index (HzI) was estimated from the hazard index of gasoline station workers' exposure to BTEX$_{emiss}$ at a distance from the fuel vapor point source. It was considered that there were potentially adverse non-carcinogenic effects of concern based on distance from source along the horizontal plane in the wind direction in meters (x). The criteria of HzI was divided by the 95[th] percentile of HzI. A HzI result of <0.5 was considered to be "a score of 1", a HzI of 0.5–1.0 was considered to be "a score of 2", a HzI of >1–2 was considered to be "a score of 3", and a HzI of >2 of HzI was considered to be "a score of 4".

Concentrations of VOCs were converted to BTEX$_{emiss}$ concentrations by using gasoline emission factors of EPA (AP-42) [25]. The BTEX$_{emiss}$ concentration was calculated according to the following equation; BTEX$_{emiss}$ = (VOCs x EF$_{BTEX}$)/120 ppm, where BTEX$_{emiss}$ represents benzene, toluene, ethylbenzene, and xylene concentrations (ppm), VOCs is the concentration from a flammable gas detector (ppm), EF$_{BTEX}$ is the gasoline emission factors of benzene, toluene, ethylbenzene, and xylene, which were 7.7, 70.7, 14.2 and 70.7 ppb, respectively.

The BTEX$_{emiss}$ concentrations at different distances were estimated according to exposure distance by an air pollution model; the Gaussian dispersion model detailed below;

$$c(x,y,z) = \frac{Q}{2\pi\mu\sigma_y\sigma_z} \lfloor exp - \left(\frac{y^2}{2\sigma_y{}^2}\right) \rfloor exp\left\{\lceil\frac{-(z-H)^2}{2\sigma_z{}^2}\rceil + \lceil\frac{-(z+H)^2}{2\sigma_z{}^2}\rceil\right\}$$

where C (x,y,z) represents the BTEX$_{emiss}$ concentration based on exposure distance (mg/m$^3$), Q represents the emission rate from source (average 0.17 g/s); $\sigma_y$ represents the dispersion coefficient of BTEX$_{emiss}$ on the x axis in meters (0.15–10 meters in distance); $\sigma_z$ represents the dispersion coefficient of BTEX$_{emiss}$ on the y axis in meters (1.5 meters; breathing zone); μ represents a velocity of wind at a source above ground level (average 10 m/s); H represents the net high plume: h$_s$+Δh$_i$ (meters), where h$_s$ represents the air pollution source of the high plume, and h$_i$ represents the plume rise, the plume high after emission; x represents a distance from the source along the horizontal plane in the wind direction (meters); y represents a distance from the source along the level perpendicular to the wind direction (meters); and z represents the height of the source from the ground (breathing zone = 1.5 meters).

The possibility (P) of fire ignition was determined according to the flammability limit (percentage of the Lower Explosive Limit and Upper Explosive Limit (%LEL-UEL)). It was divided

into two groups, which were the out-of-range and in-range %LEL-UEL of flammable gas and vapor. The criteria of the flammability level were divided by the 95th percentile of values to be compared to the range of 1.3–7.4% as specified in the fire ignition or explosion range [26]. A result outside the range of %LEL-UEL was considered "a score of 1" and a result in the range of %LEL-UEL was considered "a score of 2".

## Statistical analysis

This work was analyzed by using STATA version 10 software. Descriptive statistics were used for gasoline station characteristics and health risk assessment. The Chi-squared test was done for the association of factors with the hazard index. The statistical significance was identified at a *p-value* of $\leq 0.05$.

## Results

### Characteristics of gasoline stations

There were 10.6% of gasoline stations located in urban areas, 80.9% in suburban areas and 8.5% in rural areas. A total of 27 (57.5%) stations were open 16 hours per day (06.00 am—22.00 pm) and 20 (42.6%) stations were open 24 hours for service. There was an average of 23 ±12 fuel nozzles per station (min: max = 8:48) and there were no VRS installed on the dispensers. There was an average safe distance of 20.4 meters (min: max = 6:45) between the fuel dispenser and service building, while five stations (10.6%) had a safe distance of less than 8 meters.

The average daily amount of gasoline sold was 3,382.8±2,382.9 liters. The gasoline stations were classified according to service type; 14 stations (29.8%) were type IV (fuel dispenser house, oil storage tanks, office, maintenance store, minimart, coffee shop, and food court), 21 stations (44.7%) were type III (fuel dispenser house, oil storage tanks, office, maintenance store, minimart, and coffee shop), six stations (12.8%) were type II (fuel dispenser house, oil storage tanks, office, and maintenance store) and six stations (12.8%) were type I (fuel dispenser house, oil storage tanks, and office).

The mean concentration of VOCs while refueling was 410.0±172.0 ppm (min: max = 158:810). There were 43 stations (91.5%), which had a VOC emission of more than 200 ppm. The recorded wind velocity average on-site was 10 meters/min.

### BTEX concentrations and health hazard index in fuel dispenser areas

The resulting maximum BTEX concentrations measured by active air sampling were 136.9, 406.0, 24.1, and 105.5 ppb for benzene, toluene, ethylbenzene, and xylene, respectively. The benzene concentration of two stations (4.3%) exceeded the safety standard of 100 ppb, or 320 μg/m³ [19]. The health risk of adverse non-cancer effects (HI) ranged from <0.1 to 7.3, and five gasoline stations (10.6%) reached an unacceptable level of risk (HI>1) with regard to BTEX exposure, as shown in Table 2.

Regarding the correlation between the health risk based on chronic inhalation and the risk factors, the level of risk found in stations with higher amounts of daily gasoline sold (≥3,000 liters/day) was significantly higher than in those with lower volumes of gasoline sold. Likewise, higher numbers of gasoline dispensers resulted in a significantly higher risk than smaller numbers of nozzles.

When analyzing the potential inhalation intake based on the 95th percentile of BTEX concentration, according to the location (urban, suburban, rural) of the gasoline station, it was shown that the HI across all zones indicated an unacceptable risk (HI>1) at five stations which

**Table 2. BTEX concentrations and hazard index at the dispenser area (N = 47).**

| Parameter | Air concentration | | | HQ | |
|---|---|---|---|---|---|
| | µg/m³ mean(min-max) | ppb (min-max) | >TWA; n(%) | min-max | >1 n(%) |
| Benzene(B) | 33.1(0.1–437.5) | 0.1–136.9 | 2(4.3) | 0.1–5.5 | 4(8.5) |
| Toluene(T) | 142.7(30.5–1,529.7) | 8.1–406.0 | 0 | <0.1–0.1 | 0 |
| Ethylbenzene(E) | 14.4(3.4–104.9) | 0.8–24.1 | 0 | <0.1–0.1 | 0 |
| Xylene(X) | 41.3(1.7–458.3) | 0.4–105.5 | 0 | <0.1–1.7 | 1(2.1) |
| HI$_{BTEX}$ | | | | <0.1–7.3 | 5(10.6) |

were in urban and suburban areas. There was an almost significant difference between the HI of the three zones. The service type of stations and amount of gasoline dispensers and VOC emissions led to cumulative effects on the upward trend of the hazard index. Five stations classified as type III+IV gasoline stations reached an unacceptable level of risk (HI>1). A total of more than 12 gasoline dispensers (including gasoline 95, octane 91, and octane 95) increased the impact on workers' health. Furthermore, there was also concern that VOC emissions from dispensers (≥200 ppm) could raise health risk potential, as shown in Table 3.

## Health hazardous zone (HHZ) and fire hazardous zone (FHZ)

The mean BTEX$_{emiss}$ values converted from VOCs were used to analyze the hazardous exposure index (HzI) and estimate the health hazardous zone (HHZ). It was found that two stations (4.3%) had a HzI value of more than 1 at a distance of 2 meters, with a value of 1.2 at the 95th percentile of HzI (maxHzI = 2.7). One station had been shown to have a HzI value between 0.5

**Table 3. The factors correlating with the hazard index (HI) of workers' exposure to BTEX at gasoline stations (N = 47).**

| Parameters | HI of BTEX | | p-value |
|---|---|---|---|
| | HI≤1, n(%) | HI>1, n(%) | |
| Location of gasoline station | | | 0.06 |
| Rural (n = 4) | 4(100.0) | 0 | |
| Suburban (n = 38) | 34(89.5) | 4(10.5) | |
| Urban (n = 5) | 4(80.0) | 1(20.0) | |
| Daily gasoline sold (liters) | | | 0.02* |
| <3,000 (n = 23) | 23(100.0) | 0 | |
| ≥3,000 (n = 24) | 19(79.2) | 5(20.8) | |
| Service-zone type of station | | | 0.16 |
| Type I+II (n = 12) | 12(100.0) | 0 | |
| Type III+IV (n = 35) | 30(85.2) | 5(14.3) | |
| Number of gasoline dispensers | | | 0.05* |
| <12 (n = 19) | 19(100.0) | 0 | |
| ≥12 (n = 28) | 23(82.1) | 5(17.9) | |
| Number of fuel nozzles | | | 0.20 |
| <12 (n = 8) | 8(100.0) | 0 | |
| ≥12 (n = 39) | 34(87.2) | 5(12.8) | |
| VOC concentration (ppm) | | | 0.47 |
| <200 (n = 4) | 4(100.0) | 0 | |
| ≥200 (n = 43) | 26(88.4) | 4(11.7) | |

*Significant at p-value ≤ 0.05

to 1 at a distance of 8 meters, with a maximum HzI value of 0.53. From this finding, the area around the dispenser was divided into two health hazardous zones (HHZ), i.e. within a two-meter radius of the dispenser was classified as health hazardous zone-I (HHZ-I) and the area >2–8 meters from the dispenser was classified as health hazardous zone-II (HHZ-II), as shown in Table 4.

The fire hazardous zones (FHZ) were decided according to the 95th percentile of % LEL-UEL. A result outside the range of 1.3–7.4% was considered "fire hazardous zone-II; FHZ-II" and a result in the range of 1.3–7.4% was considered "fire hazardous zone-I; FHZ-I". The results were out of the range of the flammability limit level (less than the 1.3% within a two-meter radius of the dispenser area). Therefore, fire hazardous zone-I was classified as within a two-meter radius of the dispenser and fire hazardous zone-II was classified as the area >2 meters to 8 meters from the dispenser, as shown in Table 4.

### Hazardous area classification

Hazardous area (HZ) classification was determined from the results of the matrix multipliers as presented in Table 5 for "hazardous area-I: HZ-I" and "hazardous area-II: HZ-II". As shown by the data, it was found that two stations (4.26%) out of 47 stations scored higher than 3 at up to 4 meters from the dispenser; therefore, hazardous area-I was designated as up to 4 meters from the dispenser. Based on a score of 1–2 and the safety action value of the hazardous exposure index (HzI = 0.5 to 1), hazardous area-II was initially determined as up to 8 meters from the dispenser. However, it was eventually determined as >4–8 meters because the safety action set of HzI values was found to be 0.56 at 8 meters radius from the dispenser. The graphic representation of hazardous areas discovered in this study for safety action is shown in Fig 1.

### Discussion

BTEX concentrations in the ambient air of working surroundings throughout the period spent at dispensing areas were lower than in previous studies in Bangkok, Thailand [11], but benzene and toluene concentrations were higher than those of other countries [13, 14] and in excess of the NIOSH standard level (100 ppb), like in the study of gasoline station workers in Kuwait [15]. A study of a gasoline station in Malaysia [9] found that concentrations were quite lower or close to the range in our study. Moreover, it was found that the benzene concentration in the dispenser area of two stations exceeded the safety standard and was higher than the previous finding in Khon Kaen, Thailand [12]. Our previous report at the fuel storage tank areas of gasoline stations [6] showed that they had higher benzene concentrations than the dispensing

**Table 4. Hazardous exposure index and fire possibility calculated for distances (in meters) around the dispensers at gasoline stations (N = 47).**

| Distance (meter) | Hazardous exposure index (HzI) | | | | | %LEL-UEL(1.3–7.4) | | | |
|---|---|---|---|---|---|---|---|---|---|
| | <0.5 n(%) | 0.5 to 1 n(%) | >1 n(%) | 95th HzI | max HzI | Out-range n(%) | In-range n(%) | 95th | max |
| 0.15 | 11(23.4) | 18(38.3) | 18(38.3) | 22.7 | 27.9 | 0(0) | 47(100) | 21.8 | 26.0 |
| 1 | 39(82.9) | 0 | 8(17.0) | 4.9 | 12.6 | 24(51.0) | 23(49.0) | 6.5 | 7.8 |
| *2* | 45(95.7) | 0 | *2(4.3)* | *1.2* | *2.7* | 44(93.6) | *3(6.4)* | *2.0* | *2.4* |
| 3 | 45(95.7) | 2(4.3) | 0 | 0.9 | 0.9 | 47(100) | 0 | 0.6 | 0.8 |
| 5 | 45(95.7) | 2(4.3) | 0 | 0.4 | 0.8 | 47(100) | 0 | 0.1 | 0.1 |
| *8* | 46(97.8) | *1(2.1)* | 0 | 0.2 | *0.5* | 47(100) | 0 | <0.1 | <0.1 |
| 10 | 47(100) | 0 | 0 | <0.1 | 0.1 | 47(100) | 0 | <0.1 | <0.1 |

Italic and bold numbers indicate the distance in meters according to a HzI>1 (zone-I) and 0.5 to 1 (zone-II); or an in-range %LEL-UEL.

Table 5. Hazardous area classification by matrix multipliers of HzI and %LEL-UEL at the gasoline stations (N = 47).

| Distance (meter) | HzI | | %LEL-UEL | | Hazardous area (HZ) | |
|---|---|---|---|---|---|---|
| | 95th | Max | 95th | Max | HZ-I;≥3 n(%) | HZ-II;1–2 n(%) |
| 0.15 | 22.7 | 27.9 | 21.8 | 26 | 47(100) | 0(0) |
| 1 | 4.8 | 12.6 | 6.5 | 7.8 | 8(17.0) | 39(82.9) |
| 2 | 1.2 | 2.7 | 2.0 | 2.4 | 2(4.3) | 45(95.7) |
| 3 | 0.9 | 0.9 | 0.6 | 0.8 | 2(4.3) | 45(95.7) |
| *4* | 0.8 | 0.9 | 0.2 | 0.3 | *2(4.2)* | 45(95.7) |
| 5 | 0.4 | 0.8 | <0.1 | 0.1 | 0 | 47(100) |
| 6 | 0.4 | 0.7 | <0.1 | <0.1 | 0 | 47(100) |
| 7 | 0.3 | 0.6 | <0.1 | <0.1 | 0 | 47(100) |
| *8* | 0.2 | *0.5* | <0.1 | <0.1 | 0 | 47(100) |
| 9 | 0.2 | 0.4 | <0.1 | <0.1 | 0 | 47(100) |
| 10 | <0.1 | 0.1 | <0.1 | <0.1 | 0 | 47(100) |

areas, which explains that the fuel loading amount was related to benzene exposure, resulting in consequential adverse effects on the health of workers.

From the adverse-effect risk assessment findings on exposure to BTEX via inhaled air in working zones, it was found that there were five stations which had a high health risk value exceeding an acceptable limit (HI>1) for BTEX exposure. As a result, benzene exposure at five gasoline stations was assessed to be unacceptable with regard to human risk, which was supported by recent studies in Thailand [6] and Malaysia [9, 10], which has similar climatic conditions to Thailand. The previous findings showed that an unacceptable risk (HQ>1) of benzene exposure at gasoline stations was related to their locations, i.e. there were higher risks in urban and suburban areas in comparison to rural areas [5], as confirmed by this study, and the characteristics of the refueling service function, i.e. working close to the gasoline dispensers, could lead to a higher health risk [5].

The assessed human health risk of BTEX exposure in the dispenser areas was significantly correlated with the daily amount of gasoline sold and the number of benzene dispensers. These factors potentially caused the hazard index value to be higher than 1 (HI>1). In particular, with regard to the amount of benzene sold, this study supports previous findings in that

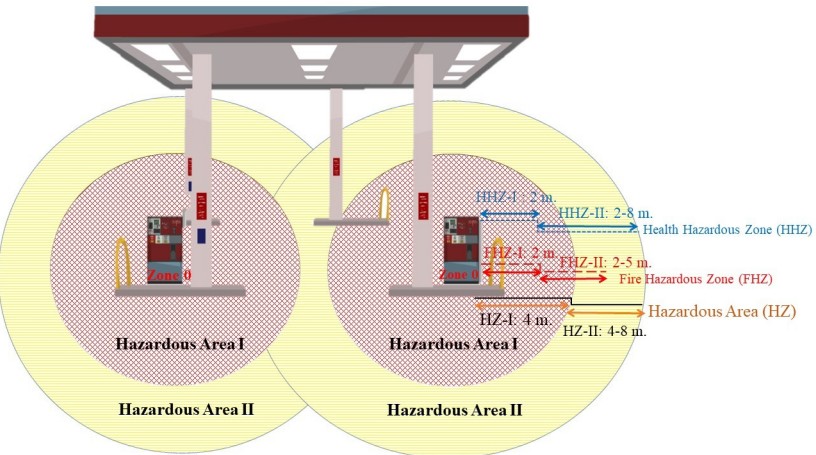

Fig 1. Hazardous area-I and area-II classification (HZ) to cover HHZ and FHZ.

high levels of service provision were related to higher exposure to benzene among gasoline workers [12]. Concerning the location of gasoline stations, chronic inhalation of the benzene concentrations at suburban stations resulted in a significantly higher risk than in other zones. Moreover, the gasoline station workers' health impact was related to high VOC concentrations due to exposure during the refueling of vehicles, as shown in a previous report [5].

The findings regarding the hazardous area around fuel dispensers meant that hazardous area-I was classified as up to 4 meters in radius, and area-II was classified as up to 8 meters in radius. The hazardous area, which resulted from a risk matrix which considered appropriate concentrations of flammable gases for fire ignition and exposure to volatile organic chemicals, could meet the specifications of the defined hazardous zone of the IEC international standard [27]. According to the hazardous zones defined by the 2013 ministerial regulations in Thailand [28], zone-I is only 1 meter in radius, and zone-II is classified as 1–1.5 meters around the dispensers for safety control of fire risk. However, our study found that the hazardous areas of fire risk and health risk of workers cover an area up to 8 meters in radius around dispensers, as shown in Fig 1.

Hazardous area-I had a high potential of workers' health risk, which was supported by the previous study of urinary tt-MA biomarker detection of benzene exposure in those working close to fuel nozzle during refueling [29]. This finding seems to be consistent and covered zones wider than the 1.5-meter radius around the dispensers of the previous study, which found that fire risk zone I had an intolerable risk in the fire risk assessment [30]. Moreover, benzene exposure was previously reported as higher in refueling workers than in those doing other jobs at gasoline service stations [5, 29].

In addition, this study found that five gasoline stations had service types which included convenience stores, minimarts, parking lots, and facilities located in hazardous area-II (up to 8 meters in radius from the dispensers), which meant that they had a hazard index higher than 0.5 (safety action point). So, entrepreneurs must strictly control hazards harmful to the health of gasoline station clients and working attendants.

## Conclusions and recommendations

BTEX concentrations in the working ambient air of gasoline stations were in the range of 0.1–136.9, 8.1–406.0, 0.8–24.1 and 0.4–105.5 ppb for benzene, toluene, ethylbenzene, and xylene, respectively. There were values which exceeded the safety standard value for benzene concentration. Risk assessment of workers on BTEX exposure indicated that six stations had reached an unacceptable level of risk (HI>1), which was related to the number of gasoline dispensers and the amount of daily gasoline sold. Hazardous area-I was classified as up to 4 meters in radius around the dispenser. The suggestion is that entrepreneurs must strictly control safety operation practice methods, such as wearing personal protective equipment, and provide safety training for workers to raise awareness of protecting against BTEX exposure in addition to a health surveillance program. Above all, entrepreneurs should control BTEX vaporization by installing VRS on dispenser nozzles, and make the hazardous areas clear by marking a safety line and informing clients that hazardous area-II is up to 8 meters in radius around the dispensers.

## Supporting information

**S1 File. Survey form English.**
(PDF)

**S2 File. Survey form Thai.**
(PDF)

## Author Contributions

**Conceptualization:** Sunisa Chaiklieng.

**Data curation:** Sunisa Chaiklieng.

**Investigation:** Sunisa Chaiklieng.

**Methodology:** Sunisa Chaiklieng.

**Writing – original draft:** Sunisa Chaiklieng.

**Writing – review & editing:** Sunisa Chaiklieng.

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
