## [Decision Letter · Decision Letter 0]

21 Jan 2021

PONE-D-20-27491

Risk assessment on workers’ exposure to BTEX and hazardous area classification at gasoline stations

PLOS ONE

Dear Dr. Chaiklieng,

Thank you for submitting your manuscript to PLOS ONE. After careful consideration, we feel that it has merit but does not fully meet PLOS ONE’s publication criteria as it currently stands. Therefore, we invite you to submit a revised version of the manuscript that addresses the points raised during the review process.

Thank you for your manuscript. I agree with the comments provided by the reviewers, and consider that the manuscript needs major revision. Please could you consider them and submit a revised version addressing the comments and feedback provided by the reviewers and by myself?

We look forward to receiving your revised manuscript.

Kind regards,

Antonio Peña-Fernández, PhD

Academic Editor

PLOS ONE

Journal Requirements:

Furthermore, please provide additional details regarding participant consent. In the ethics statement in the Methods and online submission information, please ensure that you have specified (1) whether consent was suitably informed and (2) what type you obtained (for instance, written or verbal). If your study included minors under age 18, state whether you obtained consent from parents or guardians. If the need for consent was waived by the ethics committee, please include this information

3.Thank you for stating the following in the Acknowledgments Section of your manuscript:

"This study was financially supported by the National Research Council of Thailand (NRCT6100007)."

"The author(s) received no specific funding for this work"

Additional Editor Comments:

Dear author,

Thank you for your manuscript. I agree with the comments provided by the reviewers, and consider that the manuscript needs major revision. The manuscript should be carefully reviewed, re-written and improved for consideration for publication. The revised version should also provide more information (and updated) on the toxicological aspects of BTEX compounds. More information is also required about on the quality controls undertaken in the laboratory for the analysis of the samples. This section needs comprehensive revision. Thus, Author should explain the quality controls undertaken for the quantification of BTEX in the samples, and the reference materials used or quality controls, so the results are reliable. The author should explain what ppbs stands for, i.e. are they in µg/m3?, and indicate the limits of detection for each substance. All the measures were higher than the LoD? The author should indicate when the samples were collected. I agree with the comments provided by the second reviewer, that author should compare the data found with other similar studies performed in their country and in the recent literature. Lines 71-72 should be rewritten for clarity.

Best wishes

Antonio

Reviewers' comments:

Reviewer's Responses to Questions

**Comments to the Author**

1. Is the manuscript technically sound, and do the data support the conclusions?

Reviewer #1: Yes

Reviewer #2: Yes

2. Has the statistical analysis been performed appropriately and rigorously? 

Reviewer #1: Yes

Reviewer #2: Yes

3. Have the authors made all data underlying the findings in their manuscript fully available?

Reviewer #1: Yes

Reviewer #2: Yes

4. Is the manuscript presented in an intelligible fashion and written in standard English?

Reviewer #1: Yes

Reviewer #2: Yes

5. Review Comments to the Author

Reviewer #1: This is a relevant study on workers exposure to BTEX, a worlwide problem. The study describes well the methodology followed and how the exposure and risk have been estimated. Results are well described. The authors may consider to expand on possible solutions or exposure controls to prevent workers exposure.

Specific comments:

Title: Risk assessment "of" workers´s exposure (delete "on").

Lines 31-32. "radius around the fuel dispensers" (delete "in")

Line 93-94: "as described in NIOSH method 1501 [15]" delete "which followed NIOSH number 1501 [15] methodology

Line 95: "monitored" not "monitoring"

Line 103: describe the acronym LEL-UEL

Line 171-176: the terms Ox, Oy are not represented in the formula. It is not clear what terms in the formula the refer to.

Reviewer #2: General comment

The manuscript discusses the risk assessment of workers’ exposure to BTEX and hazardous area classification at gasoline stations in Thailand. The results show the concentration of Benzene exceeds the limit of 100 ppb. Risk assessment of workers on BTEX exposure indicated that six stations had reached an unacceptable risk (HI>1) which was related to the number of gasoline dispensers and the amount of daily gasoline sold. Overall the study is interesting. Nevertheless, the author still needs to improve the writing with the inclusion of references to strengthen the introduction of the manuscript. The methodology needs to be improved with information on sampling and QA/QC.

Detail Comment

1. Abstract: Include the full name of EPA-IRIS

2. Safe standard of benzene (100 ppb): Include name of the agency on country suggest the value of concentration.

3. In term of comparison with previous studies, I suggested the author include previous similar studies conducted in Thailand such as:

Dacherngkhao, T., Chaiklieng, S., 2019. Risk assessment on BTEX exposure at fuel storage tank area in the gasoline station. Indian Journal of Public Health Research and Development 10, 2281-2286.

Kitwattanavong, M., Prueksasit, T., Morknoy, D., Tunsaringkarn, T., Siriwong, W., 2013. Health Risk Assessment of Petrol Station Workers in the Inner City of Bangkok, Thailand, to the Exposure to BTEX and Carbonyl Compounds by Inhalation. Human and Ecological Risk Assessment 19, 1424-1439.

Tunsaringkarn, T., Siriwong, W., Rungsiyothin, A., Nopparatbundit, S., 2012. Occupational exposure of gasoline station workers to BTEX compounds in Bangkok, Thailand. International Journal of Occupational and Environmental Medicine 3, 117-125.

The authors also compared other similar studies on BTEX from motor vehicles conducted in Thailand.

4. Interm comparison with other countries I suggest the author to consider similar studies conducted in Southeast Asian countries which have similar weather condition such as:

Latif, M.T., Abd Hamid, H.H., Ahamad, F., Khan, M.F., Mohd Nadzir, M.S., Othman, M., Sahani, M., Abdul Wahab, M.I., Mohamad, N., Uning, R., Poh, S.C., Fadzil, M.F., Sentian, J., Tahir, N.M., 2019. BTEX compositions and its potential health impacts in Malaysia. Chemosphere 237.

5. Provide detail information on BTEX sampling procedures. What is the specific time for 4 hours of BTEX measurement? Provide information on replications and QA/QC including the method detection limit for BTEX measurement.

6. The characteristics of sampling stations can be included in the Methodology section rather than the Results section.

7. Compare the concentration of BTEX record in this study to previous similar studies.

6. PLOS authors have the option to publish the peer review history of their article (what does this mean?). If published, this will include your full peer review and any attached files.

Reviewer #1: No

Reviewer #2: No

---

## [Author Response · Author response to Decision Letter 0]

21 Mar 2021

PONE-D-20-27491

Risk assessment on workers’ exposure to BTEX and hazardous area classification at gasoline stations

Dear PLOS ONE Academic Editor,

According to the comments of reviewers and editor, we author appreciated all suggestions to improve the paper presentation and all correction and clarifications are the following;

Reviewer Comments Answers and additional comments Location

Academic Editor Ensure that your manuscript meets PLOS ONE's style requirements, including those for file naming Format was rechecked and approved 

 Include additional information regarding the survey or questionnaire used in the study and ensure that you have provided sufficient details that others could replicate the analyses. Questionnaire was included to one uploaded attached file; Gasoline station data survey form 

 Remove any funding-related text from the manuscript Financial disclosure was deleted from an acknowledgement, and updated the funding statement 

 provide more information (and updated) on the toxicological aspects of BTEX compounds Added more referenced of Thai gasoline workers exceeded the acceptable limit [5,7,8] and similar to the studies in Malaysia [9, 10] on comment of reviewer #2 

 required about on the quality controls undertaken in the laboratory for the analysis of the samples QC and analysis of samples were edited to provide more information in the methodology section. Data collection;

A flammable gas detector was used to measure the total VOCs emission (Photo ionization detector (PID) sensor (detection range of 0 to 1000 ppm) and the lower explosive limit - upper explosive limit (LEL-UEL) with Non-Dispersive Infrared (NDIR) combustible sensors (range of 0-100%LEL)

 when the samples were collected. Sample collected in the season was stated Data collection;

The samples were collected between June and July 2018 June and July 2018, at times when the temperature was 28.08 ±1.58 °C, humidity was 78.51±8.89%, and wind velocity was 10.0 ±2.82 m/min

Reviewer #1 1. Title: Risk assessment "of" workers´ exposure (delete "on"). Revised; Title: Risk assessment "of" workers´ exposure (delete "on"). Title ..of…

 2. Lines 31-32. "radius around the fuel dispensers" (delete "in") Revised; “radius around the fuel dispensers” Abstract;

The risk matrix classified …radius around the fuel dispensers

 3. Line 93-94: "as described in NIOSH method 1501 [15]" delete "which followed NIOSH number 1501 [15] methodology Revised; "as described in NIOSH method 1501 [15]" Data collection;

The BTEX concentrations were measured… as described in NIOSH method 1501…

 4. Line 95: "monitored" not "monitoring" Revised; "monitored" Data collection;

The exposure to BTEX was monitored

 5. Line 103: describe the acronym LEL-UEL Revised; “describe the acronym LEL-UEL “ Data collection;

A flammable gas detector … lower explosive limit - upper explosive limit (LEL-UEL)…

 6. Line 171-176: the terms Ox, Oy are not represented in the formula. It is not clear what terms in the formula the refer to. Revised; Changed the symbol “σy, σz” to match the equation. Hazardous area classification at gasoline stations;

The BTEXemiss concentrations at different distances were σy σz ..

Reviewer #2 1. Abstract: Include the full name of EPA-IRIS Revised; Added full name of EPA-IRIS in abstract (however, it caused abstract to be more than 250 word). Abstract;

The risk assessment of gasoline workers at … US. Environmental Protection Agency-IRIS

 2. Safe standard of benzene (100 ppb): Include name of the agency on country suggest the value of concentration. Revised; “NIOSH exposure limits of 100 ppb benzene concentration” Abstract;

Results showed that the BTEX concentrations… of the NIOSH exposure limit of 100 ppb of benzene concentration.

 3. In term of comparison with previous studies, I suggested the author include previous similar studies conducted in Thailand Revised; comparison with previous studies in introduction and discussion. Introduction; 

It was also shown that the lifetime cancer risk…8] 

Discussion; 

…and was higher than the previous finding in Khon Kaen, Thailand [12]

 4. In term comparison with other countries I suggest the author to consider similar studies conducted in Southeast Asian countries which have similar weather condition Revised; comparison with previous studies in Southeast Asian countries in introduction and discussion. Introduction; 

It was also shown that the lifetime cancer risk… and similar to the studies in Malaysia [9, 10].

Discussion; 

.. like in gasoline station of Kuwait study [15]. The finding of concentration at gasoline station in Malaysia [9] was quite lower or closed to the range in our study.

 5. Provide detail information on BTEX sampling procedures. What is the specific time for 4 hours of BTEX measurement? Provide information on replications and QA/QC including the method detection limit for BTEX measurement. Revised; 

1. the information of 4 hours measurement was calculated to be 8 hour exposure to compare to the standard limit of working period, which were added to the method.

2. Added the limit of detection (LOD) of all BTEX

and the VOCs measured by gas detector 

 Data collection;

1. The exposure to BTEX was monitored… covered the high service hours for 4 hours sampling.

2. Each tube was extracted with… and limit of detection (LOD) was < 0.001 ppm and < 0.003 ppb for toluene and benzene and < 0.05 ppb for ethyl benzene and xylene.

……measure the total VOCs emission (Photo ionization detector (PID) sensor (detection range of 0 to 1000 ppm) and the lower explosive limit - upper explosive limit (LEL-UEL) with Non-Dispersive Infrared (NDIR) combustible sensors (range of 0-100%LEL).

 6. The characteristics of sampling stations can be included in the Methodology section rather than the Results section. Revised; 

- Added sample inclusion criteria; gasoline dispenser nozzles, and gasoline stations operation time in the methodology section. Sample size; 

There were two inclusion criteria… and gasoline dispenser nozzles; and 3) they had to have more than 8 hours of daily operation time.

 7. Compare the concentration of BTEX record in this study to previous similar studies

 Revised; 

1. BTEX comparison on in discussion section.

2. HQ and HI comparison in discussion Discussion

1. BTEX concentrations in the ambient… and accessed of the NIOSH standard level (100 ppb) like in gasoline station of Kuwait study [15]. The finding of concentration at gasoline station in Malaysia [9] was quite lower or closed to the range in our study.

2. From the adverse effect risk… in Thailand [6] and Malaysia [9, 10], where had the similar climatic conditions to Thailand. The study showed that… study [5] as confirmed by this study and the characteristic of a function as refueling service working closed to the gasoline dispensers could exhibit the higher health risk [5].

---

## [Editor Report · Decision Letter 1]

29 Mar 2021

Risk assessment of workers’ exposure to BTEX and hazardous area classification at gasoline stations

PONE-D-20-27491R1

Dear Dr. Chaiklieng,

We’re pleased to inform you that your manuscript has been judged scientifically suitable for publication and will be formally accepted for publication once it meets all outstanding technical requirements.

Kind regards,

Antonio Peña-Fernández, PhD

Academic Editor

PLOS ONE

Additional Editor Comments (optional):

Thank you for addressing the different comments provided and congratulations on your manuscript.
---

## [Editor Report · Acceptance letter]

6 Apr 2021

PONE-D-20-27491R1 

Risk assessment of workers’ exposure to BTEX and hazardous area classification at gasoline stations 

Dear Dr. Chaiklieng:

I'm pleased to inform you that your manuscript has been deemed suitable for publication in PLOS ONE. Congratulations! Your manuscript is now with our production department. 

Kind regards, 

on behalf of

Dr. Antonio Peña-Fernández 

Academic Editor

PLOS ONE